# Spatial Reasoning Network for Zero-shot Constrained Scene Generation

## Abstract

Constrained scene generation (CSG) generates images satisfying a given set of constraints. Zero-shot CSG generates images satisfying constraints not presented in the training set without retraining. Recent neural-based models generate images with excellent details, but largely cannot satisfy constraints, especially in complex scenes involving multiple objects. Such difficulty is due to the lack of effective approaches combining low-level visual element generation with high-level spatial reasoning. We introduce a **Sp**atial **Re**asoning **N**etwork for constrained scene generation (SPREN). SPREN adds to the state-of-the-art image generation networks (for low-level visual element generation) a spatial reasoning module (for high-level spatial reasoning). The spatial reasoning module decides objects' positions following the output of a Recursive Neural Network (RNN), which is trained to learn implicit spatial knowledge, such as trees grow from the ground. During inference, explicit constraints can be enforced by a forward-checking algorithm, which blocks invalid decisions from the RNN in a zero-shot manner. In experiments, we demonstrate SPREN is able to generate images with excellent details while satisfying complex spatial constraints. SPREN also transfers good quality scene generation to unseen constraints without retraining.

## 1 Introduction

Constrained content generation has long been an important task in artificial intelligence and has many implications across domains Nauata et al. (2020); Jiang et al. (2021); Ma et al. (2021). This paper focuses on constrained scene generation – generating a realistic scene image containing multiple objects satisfying a given set of constraints. While there has been exciting progress in scene generation, especially using deep generative models Deng et al. (2021); Liu et al. (2021); Arad Hudson & Zitnick (2021), generating scenes involving multiple objects satisfying complex spatial relationships remains a challenging task. Existing approaches often cannot generate scenes which contain the right number of objects or the correct spatial relationship between the objects (according to user-defined constraints).

We hypothesize such difficulty is due to the lack of effective approaches combining low-level visual element generation with high-level spatial reasoning. From psychophysiological studies Kahneman (2011); Sowden et al. (2015); Lin & Lien (2013), human beings require multiple systems of reasoning and memory (including systems 1 and 2) to complete complex content generation tasks. Procedural (P) cognition retains the skill to generate the texture/shape of standalone objects. System 1 (S1) cognition captures "common-sense" knowledge and patterns. For example, trees are on the ground, but birds are in the sky. System 2 (S2) cognition embodies reasoning about the high level task, planning the content of the image and enforcing explicit constraints at an abstract level. Over the years, neural generative models have been very successful in learning tasks associated with P and S1-cognition, but fail consistently on S2-cognition – especially enforcing complex constraints. Traditional constraint reasoning methods are able to provide the S2-cognition necessary for our task, but they are too rigid to handle P and S1-cognition.

We introduce a **Sp**atial **Re**asoning **N**etwork for constrained scene generation (SPREN). The key idea is to add to the state-of-the-art neural generative models responsible for low-level visual element generation, or P/S1 cognition, a spatial reasoning module, which handles high-level spatial reasoning, or S2 cognition. The input of constrained scene generation is a background image and

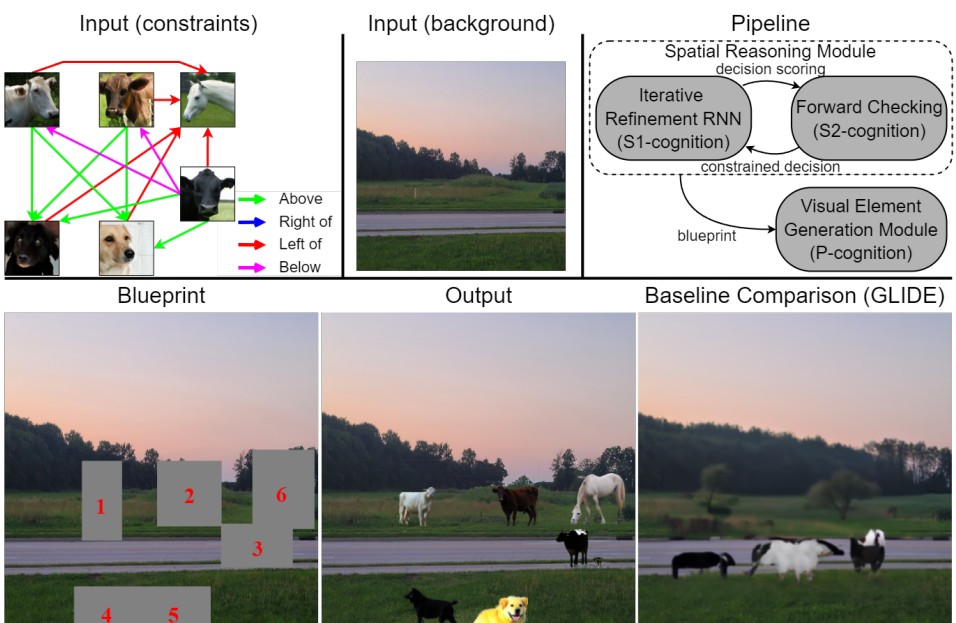

Figure 1: A scene generated by SPREN with a given background and six animals subject to complex positional constraints. The animals and the spatial constraints governing their spatial locations are shown in the upper left panel. The upper center panel shows the background image on which the scene must be placed. The upper right panel shows the high-level pipeline of our approach. The lower left panel shows the locations of these animals generated by SPREN (blueprints). The lower center panel shows a realistic image SPREN produces which satisfies constraints. The lower right panel shows a baseline scene for comparison, where constraints cannot be properly enforced.

a set of spatial constraints represented in propositional logic, and the output is the generated image containing objects satisfying constraints. The spatial reasoning module decides the objects' positions following the output of a Recursive Neural Network (RNN) in a process of iterative refinement. The RNN outputs the bounding boxes (we call them blueprints) for each object to be generated. When determining one coordinate of the bounding box, the RNN iteratively halves the range of the coordinate until it is sufficiently small. During learning, the RNN is trained to understand implicit spatial knowledge, such as trees grow from the ground and birds fly in the sky. This is done by a teacher forcing procedure which matches the bounding boxes predicted by the RNN and the ones containing the objects in the original image. During inference, explicit constraints can be enforced by a forward-checking algorithm, which blocks the decisions leading to constraint violations. The forward checking procedure also allows us to handle constraints in a zero-shot manner. During test time when novel constraints are present, the forward checking procedure blocks the output of the RNNs following the same procedure handling familiar constraints, without any retraining or fine-tuning. Figure 1 demonstrates the generative procedure of SPREN. Here, the colored arrows in the upper left panel represent the spatial constraints restricting the animals to be generated. The upper center panel represents the input background image. The lower-left panel shows the blueprint (bounding boxes) output by the spatial reasoning module. The lower-middle panel shows the final output of SPREN and the lower-right a comparison with the previous state-of-the-art GLIDE model.

In experiments, we demonstrate SPREN is able to generate images with excellent details while satisfying complex spatial constraints. We also show that SPREN works well for object aware scene generation, which is an inpainting task involving adding additional objects to an image containing existing ones satisfying constraints involving both the existing and newly added objects. SPREN also works well in zero-shot transfer learning: it generates good-quality scenes involving constraints unseen from the training set without retraining or fine-tuning. Overall, our contributions are:

- We introduce the SPREN framework for constrained scene generation, combining low-level visual element generation with high-level spatial reasoning.
- The objects are positioned satisfying explicit constraints and fit into the visual context of the image well, thanks to the spatial reasoning module.

- The final generated images contain excellent details thanks to the deep generative networks for low-level visual element generation.

- SPREN can handle zero-shot scene generation – new constraints unseen during training can be added at any time without retraining or fine-tuning.

- SPREN can successfully handle object-aware scene generation, involving handling constraints which constrain the positions of the known objects present in the background image and the newly added objects.

- SPREN admits a rich set of spatial constraints represented in propositional logic.

## 2 PROBLEM DEFINITION

### 2.1 ZERO-SHOT CONSTRAINED SCENE GENERATION (ZS-CSG)

Standard scene generation is the problem of generating a set of objects onto a given (or externally generated) background image to create a realistic scene. Constrained scene generation requires the same, but with the additional demand that objects placed in the scene satisfy a set of positional constraints. These constraints may change from scene to scene. Moreover, certain constraints may not present in the training data. At test time, the neural network needs to generate images satisfying such constraints without any fine-tuning or retraining. We refer to such use cases as zero-shot constrained scene generation. Formally, the problem can be defined as:

**Problem 1:** *(ZS-CSG) **Given**: let B be a background image, C be a set of positional constraints, and T be the set of "natural" images.*
***Find**: a scene image S with B as a background, such that all constraints in C are satisfied. Moreover, S looks "realistic"; i.e., it is visually close to the images in T.*

We use propositional logic to represent the set of positional constraints. As future work, we will attack the scene generation problem with natural language as the input. Propositional logic uses logical operators ($\land$ for "and", $\lor$ for "or", $\neg$ for "not") to connect a set of predicates. For our purposes, predicates define the names and types of objects and the spatial relations between objects. For example, *"prompt(0, a green sheep)"* means object 0 is a green sheep. *"type(0, sheep)"* defines the type of object 0 to be a sheep. *"left(0, 1, 200)"* means object 0 needs to be positioned to the left-hand side of object 1 by at least 200 pixels (the last argument is often left out, meaning object 0 is just left of 1). The semantic meaning of logic connectors remains the same as in the standard propositional logic. As an example presented in Figure 1, the constraints are shown in the upper-left panel while the input background image is in the upper-center panel. Here the colors of the arrows represent different predicates and all the predicates are connected with logic "and" ($\land$). The spatial reasoning module places the animals in the gray boxes in the lower-left panel, satisfying all the constraints. The final output image is shown in the lower-center panel of Figure 1. See the full list of predicates and logic formulations in Appendix A.

### 2.2 OBJECT-AWARE CONSTRAINED SCENE GENERATION (OA-CSG)

Aside from the ZS-CSG problem where all the objects are generated on an empty background, we also consider the more challenging object-aware constrained scene generation problem (OA-CSG). In this problem, the given background may already contain objects. There are logic constraints which specify the spatial relationships between the objects to be generated and these existing objects. This version of the problem requires that the algorithm to be aware of pre-existing objects and can take them into account during the decision-making process – both in relation to constraints and the realism of the scene.

**Problem 2:** *(OA-CSG): **Given:** let B be a background image which contains initial objects W, and C be a set of positional constraints. C may reference objects in W. T is the set of "natural" images.*
***Find** a scene image S, with B as a background, such that C is satisfied and S is "realistic"; i.e., it is visually close to the images in T.*

### 2.3 CHALLENGES

Previous approaches face major difficulties handling constrained scene generation. The ability for large diffusion models like DALL-E Ramesh et al. (2021) and GLIDE Nichol et al. (2021) to inpaint objects into scenes given a natural language prompt has made them quite attractive for constrained scene generation. A prompt can describe the objects to be generated and the constraints to be applied between those objects. However, these models consistently fail on any non-trivial instructions (such as constraining the number of objects, their types, and/or spatial relationships). Recently, a line of research encodes requirements (such as content of the images) into hidden vectors via graph or recursive neural nets Deng et al. (2021); Xu et al. (2018); Engelcke et al. (2020), and then condition image generation on these hidden vectors. These approaches provide a way to condition image generation on certain requirements, but still cannot enforce complex constraints explicitly. Further, all the conditions must be present in the training set. Otherwise, it is difficult to predict what the hidden vectors will be on an unseen new condition. Our SPREN approach, on the other hand, handles complex constraints that are unseen in the training set without retraining or fine-tuning.

### 2.4 ASPECTS OF COGNITION

Taking inspiration from the concept of *slow* and *fast* thinking Kahneman (2011), we base our approach on the idea of integrating multiple levels of reasoning for the completion of this complex task. Under this paradigm, cognition is split into system 1 "fast" thinking, composed of learned behavior that can be accessed efficiently, and system 2 "slow" thinking, which is a more intentional form of thought used in problem-solving and reasoning about abstract concepts. Our work draws the S1 and S2 cognitive aspects from this area of psychology. We also define P-cognition from the concept of procedural memory, colloquially known as muscle memory. Procedural memory requires a great deal of practice to build, but allows us to learn difficult skills through extreme repetition.

Constrained scene generation requires a synthesis of these three aspects of cognition. P-cognition involves learning the basic skills of a larger task – in this case, paining objects into a scene at a given location. S1-cognition captures fuzzy knowledge and data-driven patterns. For CSG, these are the locations of objects relative to the background and the other objects in a non-constrained case. S1 learns how to place objects "realistically". S2-cognition handles the more abstract procedures of reasoning, specifically constraint satisfaction. S2-cognition may ensure that an image "has a dog to the left and a cat to the right" but is not so concerned on the precise location of the cat and dog. In this work, S2-cognition regulates the decisions of S1-cognition, and both work together to expertly dispatch P-cognition and accomplish the scene generation.

## 3 SPATIAL REASONING NETWORK (SPREN)

SPREN is composed of one spatial reasoning module and one visual element generation module. The ***spatial reasoning module*** takes the input, and decides the spatial position of each object. In our setting, the spatial position is represented as a bounding box with parameters (*x, y, width, height*). Here, *x, y* is the coordinate of the upper-left corner and *width, height* represent the width and height of the bounding box. We call these bounding boxes "blueprints". See the lower-left panel of Figure 1 for the generated blueprints. During learning, the spatial reasoning module is presented with a set of background images and the corresponding bounding boxes of various objects in the images. The module is asked to generate bounding boxes which matches those given in the training data. This corresponds to learning implicit spatial preferences, e.g., the relative sizes of a tree compared to a house, and/or trees grow on the ground while birds flying in the sky. We implement the spatial reasoning module using an RNN-type structure, which iteratively refines the location of each object. During inference, the spatial reasoning module needs to decide the locations of each object given the background image and a set of constraints. Here, we propose a forward checking algorithm which rules out invalid outputs of the spatial reasoning module. Notice that the forward checking algorithm fits seamlessly with the RNN component and does not need to be trained. This setup allows SPREN to be transferred to new constraints not presented in the training set in a zero-shot learning setting. See Appendix D for more on our architecture and hyperparameters.

The ***visual element generation module*** takes as input the background, the prompt, and location of each object, and outputs an image patch which contains the object and can be merged seamlessly

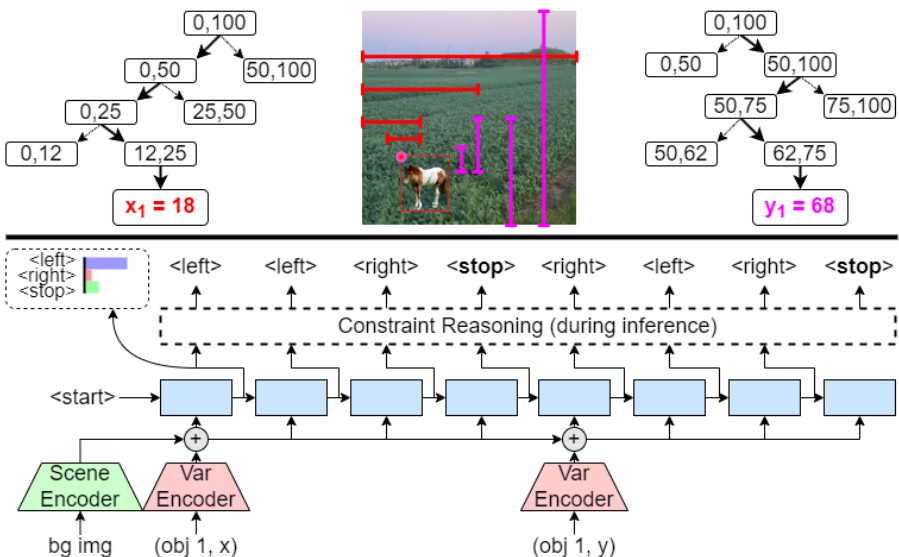

Figure 2: Rollout of the spatial reasoning module. The position of each variable is narrowed down until the stop token is selected. In the lower panel, the spatial reasoning network assigns probability scores to each decision. At the start of the scene, the hidden state is initialized from a resnet-based scene encoder encoding the background image. At the start of generating each position variable, the hidden state is added by a vector generated by the variable encoder. Constraint reasoning is applied during inference to choose a final decision which satisfies constraints. Above, the decision-making process is visualized as a tree of value ranges (to generate the $x$ and $y$ positions of the horse).

into the background image. We use state-of-the-art neural generative models as the visual element generation module. While we can train a visual element generation module ourselves, we recognize the need for huge datasets and manual effort to train these models successfully. We also would like to be able to re-use and adapt to as many state-of-the-art neural models as possible. The lower-center panel of Figure 1 demonstrates the generated image, with each bounding box filled by the visual element generation module.

## 3.1 ARCHITECTURE & SEMANTICS OF THE SPATIAL REASONING MODULE

The spatial reasoning module takes the input of the background image as well as the object identifier, and outputs the position in the form of (*x, y, width, height*) for each object. We use encoder networks to encode both the background image and the object identifier (e.g., object 0) into high-dimensional vectors. The spatial reasoning module harnesses a Recursive Neural Net (RNN) structure and outputs the four coordinates sequentially. The key idea in generating each coordinate is through *iterative refinement*. The basic unit of the RNN output is the *decision token* – a size 4 vector representing a score distribution of four possible decisions – or more accurately three possible decisions and one special start token. The input at each timestep is the previous timestep's decision as a one-hot vector. The network outputs at each step a softmax vector to assign the probability to each token. At the start of calculation, the selected variable is presumed to be within a given range (e.g. the first object's *x* value is between 0 and 100). The actions are to limit the position to be in the first half of the range (e.g. left: $0 \leq x < 50$), or in the second half of the range (e.g. right: $50 < x \leq 100$), or to terminate and select the middle value between the current minimum and maximum as the final location (e.g. stop: $x = 50$). This produces a *sequence of decisions* leading to the assignment to the corresponding variable. The sequence is embodied as a *decision string* composed of *decision tokens*. For example, the variable *x* between 0 and 100 may be assigned the decision string "left, right, stop", refining the value to $0 \leq x < 50$ as we choose the left of the range, then $25 < x < 50$ as we choose the right of the range, then $x = 37$ as we choose the center of the range. Our recursive neural network is implemented in the form of GRUs (Cho et al. (2014)). All contextual data including the background image encoding and the variable encoding is introduced into the GRU state vector (size 500) by adding to it. Figure 2 shows the processes of generating two coordinates $x_1$ and $y_1$ using the proposed neural network module, including the encoding at the start of each variable.

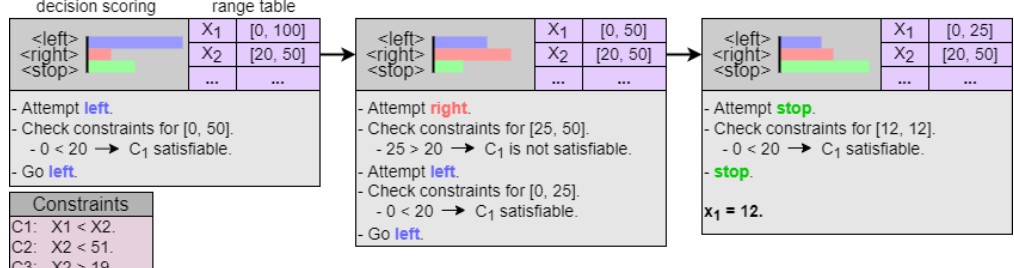

Figure 3: Example of forward checking during inference. Depth first search is applied in an order determined by sampling from a multinomial distribution of the decision scores generated by the RNN (shown here as a 3-option bar graph). The DFS is augmented with a range table, allowing for quick reference between variables and efficient recognition of un-satisfiable paths.

## 3.2 TRAINING OF THE SPATIAL REASONING MODULE

The training process uses a dataset containing background images with a bounding box for each object to be generated. Each object location is defined by the upper left point, as well as the width and height of the bounding box. These variables are converted to decision strings as described in the previous section. The training process is a supervised procedure – it trains the spatial reasoning module to generate bounding boxes (in the form of decision strings) which match those in the training dataset. This procedure teaches the neural network implicit spatial knowledge (or S1-cognition), including the relative sizes of objects (horses, dogs, trees, etc) and their common spatial relationships (birds fly in the sky, etc). We do not enforce explicit constraints during the training phase. The training of RNN is accomplished through *teacher forcing* (Lamb et al. (2016); Williams & Zipser (1989)). In teacher forcing, the ground truth decision token from the previous step drawn from the training set is used as the input to train the RNN to predict the decision of the current step. Teacher forcing can drastically speed up RNN training, but can also result in lower resilience to uncommon sequences. For this reason, a mixture of strategies – with and without teacher forcing – is used to train the model. 50% of the time the input of the RNN is set by teacher forcing and 50% set by the predictions of the RNN itself. We also randomize the order of the objects in the decision string during training. Finally, the spatial reasoning module also needs to learn to encode the background image – extracting important features like the horizon of the scene, or major obstacles. This is accomplished by back-propagating the gradients through the scene encoder network. It is difficult to find a real-world datasets of background images (containing no or few objects) and bounding boxes of the objects to be generated. We instead use real-world datasets of images containing objects and their corresponding bounding boxes (such as COCO). We remove these objects from the original images by painting over them using GLIDE. This ensures the bounding boxes of the objects are placed in their most natural positions. This step is discussed more in Appendix G.

## 3.3 INFERENCE USING THE SPATIAL REASONING MODULE

In the absence of constraints, the network can simply output a decision string for each positional variable by sampling tokens from the probability distribution outputs until a conclusive value is reached. At the beginning of a string, the start token is given as input and the variable encoder adds information to the GRU state vector. The decisions are sampled based on the probabilities output by the spatial reasoning module. For example, if the module outputs the scores $[left : 75\%, right : 5\%, stop : 20\%]$, the spatial reasoning module will select "left" 75% of the time, "right" 5% of the time, and will choose to stop 20% of the time and to choose the middle value of the current range.

If constraints are introduced, they can be enforced by limiting the decisions considered in the sampling process. A forward checking algorithm is used for this purpose. For each decision in the iterative refinement, we check that the current value range contains at least one value assignment which satisfies all constraints. If at least one constraint must be violated, we can rewind the decision string one token and resample without the option of the original choice. Our forward checking algorithm is implemented as a depth first search exploring all future expansions of the decision strings – though many algorithms could fit this role. This forward checking process is conducted after each decision made by the recursive neural network to see if constraint satisfaction is still possible. In our

previous example, we may have selected the "left" token first, but then found that no value in the remaining range for that variable can satisfy all the constraints. We resample this token with new scores $[left : 0\%, right : 20\%, stop : 80\%]$. The process is visualized in Figure 3.

The addition of the forward checking algorithm applies much-needed S2-cognition to the S1-cognition encoded within the neural net, resulting in a two part reasoning system which can intelligently plan for object placement within positional constraints.

### 3.4 THE VISUAL ELEMENT GENERATION MODULE

The blueprint from the spatial reasoning module is used to create masking boxes, which let us generate each object into the scene using the visual element generator. The result is a completed scene including all objects and adhering to all constraints. The visual element generation module is sequentially called on each box from the blueprint created by the spatial reasoning module. For the purposes of this paper, we use the reduced and filtered GLIDE diffusion model released by OpenAI. Given a masked image and a prompt, this network inpaints the masked areas accordingly. To fit the limitations of this model, all inpaint boxes must be less than 256 pixels in any dimension (this can be ensured with an extra constraint in the spatial reasoning module). The 256x256 pixel area including the box is cropped and used to inpaint the image only within the masked space. In most cases, this will maintain perfect constraint satisfaction, but sometimes GLIDE will output something that does not fit the prompt, or it will simply fail to inpaint anything except background. These inpainting failures are considered an inpaint stage constraint breach.

## 4 RELATED WORK

**Image Generation**. Realistic image generation became a feasible task along with the rise of deep neural networks. Generative autoencoders Kingma & Welling (2013); Makhzani et al. (2016), generative adversarial nets (GANs) Goodfellow et al. (2014), flow models Rezende & Mohamed (2015), and diffusion models Sohl-Dickstein et al. (2015) all do this by learning a function that transforms noise from a known distribution to a new distribution from a dataset. For example, diffusion models train the network in steps to reconstruct partial images from noise. Deep generative models are capable of producing extremely realistic images Brock et al. (2019); Karras et al. (2018); Zhu et al. (2017); Karras et al. (2019; 2020), but are limited on how controlled the generation is from image to image. For some models, conditioning Mirza & Osindero (2014) is a good strategy to increase control over generation, but this requires a very structured data source and does not guarantee the condition will be upheld.

**Image Generation from Natural Language**. A greater degree of control in image generation can be achieved by generating images from a natural language prompt. Transformers Vaswani et al. (2017) are neural nets based around multi-headed attention mechanisms. While they are most known for their ability to generate text as a language model, they have shown competence in image Chen et al. (2020) and audio Dhariwal et al. (2020) tasks as well. There now exist several deep generative models which create images from natural language interpreted by transformers Ramesh et al. (2021; 2022); Nichol et al. (2021). This is possible in part due to massive datasets extracted from the internet. While transformer-based models add control to the generation process, they often fail given complicated prompts or non-trivial constraints.

**Scene Generation**. Generating entire scenes is a difficult task even with current tools. This difficulty comes from finding realistic locations for objects within the scene which synergize with all other objects, and the internal representations that go along with that. Large transformer-based generators are capable of generating whole scenes instead of single-object images, but a single latent space limits the complexity of scenes that can be produced alone. Recurrent neural networks Engelcke et al. (2020), graph neural networks Deng et al. (2021), transformers Arad Hudson & Zitnick (2021), and energy based models Du et al. (2020); Liu et al. (2021) have had some success in filling this gap. However, standalone GAN models have been able to accomplish good scene generation as well with some modification Liu et al. (2019); Xu et al. (2018).

**Neural Networks & Reasoning**. Adding various types of reasoning to neural algorithms has been an important part of the AI research for quite some time. Much of it has focused on attempting to add "common-sense" reasoning to natural language models He et al. (2019); Trinh & Le (2018);

| Dataset | Method | Mean Pref Score % | Constraint Sat % | Training Steps | Training Time (min) |
|---|---|---|---|---|---|
| Basic | SPREN (ours) | **100** | 100 | 1600 | 90 |
| | GAN + CVX | **100** | 100 | 673 | 4 |
| Tight | SPREN (ours) | **91.7** | 100 | 2600 | 207 |
| | GAN + CVX | 81.3 | 100 | 2600 | 16 |
| Complex | SPREN (ours) | **87.5** | 100 | 2000 | 217 |
| | GAN + CVX | 72.3 | 100 | 2600 | 16 |

Table 1: Results for evaluating the spatial reasoning module. Our SPREN is able to generate object locations better satisfying the implicit preferences of object positioning in the training set (measured by the pref score) compared to baselines. Note, SPREN can also handle non-convex constraints, while "GAN+CVX" cannot. Details of this experiment are in the Appendix B.

| Method | BP Sat % | 2 Objects, 2 Constraints | | | 3 Objects, 4 Constraints | | |
|---|---|---|---|---|---|---|---|
| | | Overall Sat % | Obj Sat % | Pos Sat % | Overall Sat % | Obj Sat % | Pos Sat % |
| SPREN (ours) | **100** | **80** | **80** | **100** | **65** | **65** | **100** |
| GLIDE (center) | N/A | 5 | 5 | **100** | 0 | 0 | 0 |
| GLIDE (unbound) | N/A | 10 | 25 | 40 | 0 | 1.6 | 0 |

Table 2: Constraint satisfaction results for the full CSG experiment. SPREN outperforms GLIDE both in the blueprinting phase (generating locations) and in the final generated images when generating different number of objects subject to constraints (manually inspected). Details in the Appendix C.

Peng et al. (2015), though progress has also been made in integrating case-based reasoning once common in expert systems Im & Park (2007), as well as relational reasoning between objects or terms Santoro et al. (2017); Lamb et al. (2020); Pise et al. (2021). Very recently, constraint reasoning has been considered a fruitful domain for integration with neural nets – convex optimization Boyd & Vandenberghe (2004) as a neural net layer being a recent and applicable innovation Agrawal et al. (2019; 2020; 2021), and other constraint reasoning methods have also been successfully integrated into neural networks for a variety of applications Xue & Hoeve (2019); Bai et al. (2021); Khalil et al. (2017); Chen et al. (2021).

## 5 EXPERIMENTS

We evaluate both the spatial reasoning module alone in its ability to generate blueprints (bounding boxes for a scene), and our full SPREN model in generating constrained scenes.

**Evaluating the Spatial Reasoning Module**. We find that our approach generates blueprints that 100% satisfy spatial constraints and better match implicit preferences of object positioning compared to baselines. See table 1. Due to space restrictions, we leave all the discussions of the experimental setup, baseline description, etc, to the appendix. Note, our problem setup requires constraint satisfaction tools that can be embedded into learning the implicit preferences of object positioning (or S1 cognition). Few constraint programming tools can handle this task, except for the "GAN + CVX" approach we found Amos & Kolter (2017). However, "GAN + CVX" only handles convex objective functions and constraints, while our SPREN handles arbitrary constraints in propositional logic.

**Evaluating the Full SPREN Algorithm.** When evaluating our full CSG algorithm, SPREN surpasses the baseline both quantitatively (high percentage of constraints satisfied, see table 2) and qualitatively (high quality images, see figure 4). Due to space restrictions, we again leave the experiment setup, dataset generation, baseline descriptions, etc, to the appendix.

## 6 CONCLUSION

In this work, we considered the problem of Constrained Scene Generation (CSG), as well as the zero-shot and object-aware variants of this task. CSG represents a class of generative problems which require a tight coordination between automated reasoning and machine learning and is beyond the

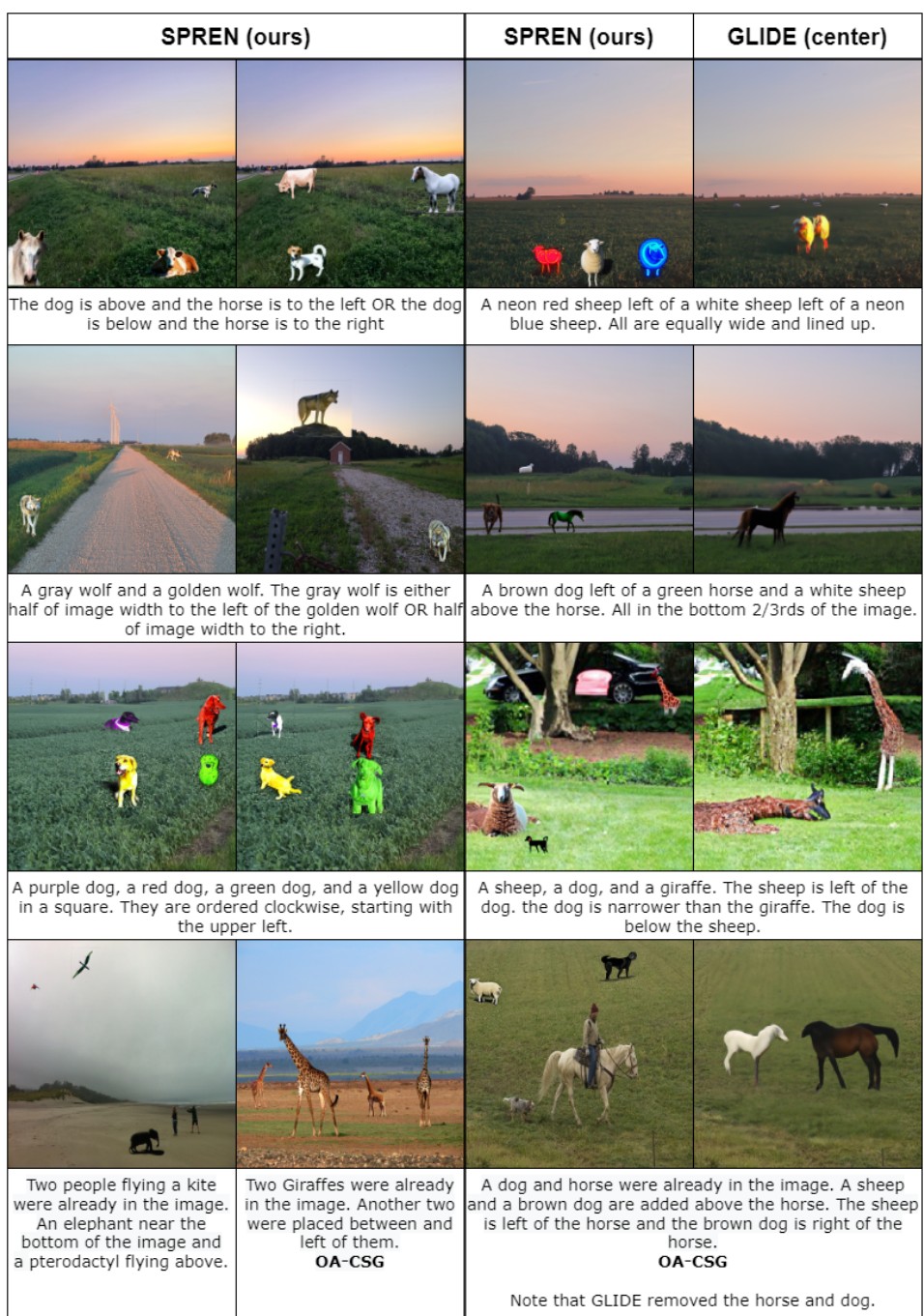

Figure 4: Selected examples of scene generation using SPREN. On the right side is a comparison between our method and GLIDE (reduced and filtered). Images for object aware constrained scene generation (OA-CSG) are noted. Constraints are described in natural language to aid readability. Full constraints in propositional logic are listed in the Appendix C.

capabilities of most neural generative models. Our proposed Spatial Reasoning Network (SPREN) combines the state-of-the-art neural generative models for low-level visual element generation with a spatial reasoning module for high-level spatial reasoning. Experiments show that SPREN is able to generate images with excellent details while satisfying complex spatial constraints. SPREN also transfers good quality scene generation to unseen constraints without retraining. Future directions of this research including expanding to other generation tasks (such as generating programs, music, etc) and taking in a richer class of inputs characterizing constraints, such as the natural language.

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

# A  PROPOSITIONAL LOGIC LANGUAGE

| Language (obj - obj) | Constraint (obj - obj) | Language (obj - val) | Constraint (obj - val) |
|---|---|---|---|
| above(a,b,c) | $y_a < y_b - c$ | above_value(a,b) | $y_a < b$ |
| below(a,b,c) | $y_a > y_b + c$ | below_value(a,b) | $y_a > b$ |
| left(a,b,c) | $x_a < x_b - c$ | left_value(a,b) | $x_a < b$ |
| right(a,b,c) | $x_a > x_b + c$ | right_value(a,b) | $x_a > b$ |
| shorter(a,b,c) | $h_a < h_b - c$ | shorter_value(a,b) | $h_a < b$ |
| taller(a,b,c) | $h_a > h_b + c$ | taller_value(a,b) | $h_a > b$ |
| narrower(a,b,c) | $w_a < w_b - c$ | narrower_value(a,b) | $w_a < b$ |
| wider(a,b,c) | $w_a > w_b + c$ | wider_value(a,b) | $w_a > b$ |
| xeq(a,b) | $x_a = x_b$ | xeq_value(a,b) | $x_a = b$ |
| yeq(a,b) | $y_a = y_b$ | yeq_value(a,b) | $y_a = b$ |
| weq(a,b) | $w_a = w_b$ | weq_value(a,b) | $w_a = b$ |
| heq(a,b) | $h_a = h_b$ | heq_value(a,b) | $h_a = b$ |

Figure 5: Definition of the propositional logic constraint language. At the top are a list of positional logic propositions on the bounding box of objects and their mathematic definitions. The point $(x, y)$ defines the upper left corner of the box, and $(x + w, y + h)$ the lower right corner. Below, "Prompt" and "Type" constraints are assignment operators specifying the prompt (given to the visual element generator) and the type (given to the spatial reasoning module). Logical operators *and* ($\wedge$), *or* ($\vee$), and *not* ($\neg$) are also defined.

Prompts and constraints are delivered in the form of a structured logical language. This language is visualized in Figure 5. Features of this language include object-to-object constraints, object-to-value constraints, assignment constraints, and a set of logical operators. Many more constraints could be implemented using this approach, but even the current ones can be used to compose complex scene structures by combining the logical operators with the positional logic propositions.

# B  SPATIAL REASONING MODULE EXPERIMENTAL SETUP

It is our aim that the spatial reasoning module be the center of integrating S1 and S2 cognition within the SPREN system, manifesting intelligent decisions based on dataset preferences and specified constraints. However, dataset preferences regarding object location are difficult to measure precisely in real world datasets like COCO. To get around this, we define 3 synthetic datasets which are generated from known distributions, allowing for evaluation against ground truth locations. In each, the underlying distribution of object locations is known to the programmer (not to the algorithm). This allows for the creation of a metric for blueprint generation – the preference score. For each generated location, if the generated value is within a range of common values, it is considered a satisfied preference. The mean number of satisfied preferences is a metric for how well the algorithm has learned the dataset preferences. See Appendix F for full specification of the datasets and preference scores.

There are very few current scene generation algorithms capable of zero-shot scene generation, which makes evaluation difficult. Despite this, SPREN planning is compared with a similar technique to ours, which also combines reasoning and deep learning, but offers a less rich set of constraints.

Specifically, we compare with a GAN augmented with a convex optimization layer Agrawal et al. (2019). Embedded CVX solvers in neural nets have been used successfully in several domains, including learning control policies Agrawal et al. (2020) and building convex optimization models Agrawal et al. (2021). Like ours, this approach can produce zero-shot constrained scenes, though the constraint set is limited to convex functions while ours is not. See Appendix E for a description of the GAN + CVX approach.

Each of the methods is trained on the three datasets until convergence. Each dataset is also matched with a set of constraints during test time. As both methods use constraint reasoning, they are both able to achieve a perfect score on this limited constraint library. Both guarantee constraint satisfaction. However, as shown in Table 1, our algorithm outperforms or matched GAN + CVX in the satisfaction score metric for every dataset.

## C  FULL SPREN ALGORITHM EXPERIMENTAL SETUP

For a more realistic test, SPREN must learn from a subset of the COCO dataset. 10,674 images each including some combination of dogs, horses, sheep, cows, elephants, bears, and giraffes. For training, foreground objects must be removed. This is done using the GLIDE (reduced and filtered) network shared by our visual element generation module to remove the known objects. This COCO background dataset and the code to reproduce it will be included in supplementary materials.

Two tests of 60 random specifications were conducted. The first randomly selects two animals from the learned set (without replacement) and generates two random constraints. The second does much the same, but with three animals and four constraints. Specifications that were not satisfiable were thrown out and re-generated (e.g. "the dog is right of the cow and the dog is left of the cow"). Random constraints included left, right, above, below, narrower, wider, taller, and shorter.

Predictably, SPREN produced perfectly satisfying outputs at the blueprint phase. However, the goal of this test is to also test the visual element generation module within SPREN. The results of these final scenes were manually checked for object satisfaction (correct number and type of objects painted in), positional satisfaction (positional constraints satisfied among scenes that have achieved object satisfaction), and overall satisfaction (correct number and type of objects painted in with all positional constraints satisfied among all scenes). We observe from Table 2 that SPREN vastly outperforms GLIDE across all metrics, and particularly when scenes become complicated (3 objects, 4 constraints).

Figure 4 shows some examples of our COCO-trained algorithm applied to custom background images. We compare this part of the algorithm with a GLIDE model inpainting the center of the given background with a text prompt similar to the structured language constraint set. GLIDE + CVX is not tested as it uses the same visual element generation module, and it is not capable of many of the provided constraints.

The following tables specifies the precise constraint language inputs used to produce each SPREN image.

| Constraint Language | Natural Language |
|---|---|
| prompt(0, a cow); type(0,cow); prompt(1, a dog); type(1,dog); prompt(2, a horse); type(0,horse); below_val(0,400,0); below_val(1,400,0); below_val(2,400,0); or( and(above(1,0,300),above(1,2,300), left(2,0,600),left(2,1,600)), and(below(1,0,300),below(1,2,300), right(2,0,600),right(2,1,600) )) | The dog is above and the horse is to the left OR the dog is below and the horse is to the right |
| prompt(0,a golden wolf) type(0,dog) prompt(1,a gray wolf) type(1,dog) or(left(0,1,500), right(0,1,500)) | A gray wolf and a golden wolf. The gray wolf is either half of image width to the left of the golden wolf OR half of the image width to the right. |
| prompt(0,a bright purple dog) type(0,dog) prompt(1,a bright red dog) type(1,dog) prompt(2,a bright yellow dog) type(2,dog) prompt(3,a bright green dog) type(3,dog) yeq(0,1) yeq(2,3) xeq(0,2) xeq(1,3) left(0,1,400) above(0,2,200) | A purple dog, a red dog, a green dog, and a yellow dog in a square. They are ordered clockwise, starting with the upper left. |
| prompt(0, a neon red sheep); type(0,sheep) prompt(1, a white sheep) type(1,sheep) prompt(2, a neon blue sheep) type(2,sheep) yeq(0,1) yeq(0,2) weq(0,1) weq(0,2) left(0,1,200) left(1,2,200) | A neon red sheep left of a white sheep left of a neon blue sheep. All are equally wide and lined up. |
| prompt(0, a brown dog) type(0,dog) prompt(1, a green horse) type(1,horse) prompt(2, a white sheep) type(2,sheep) left(0,1,200) above(2,1,300) below_value(0,300) below_value(1,300) below_value(2,300) | A brown dog left of a green horse and a white sheep above the horse. All in the bottom 2/3rds of the image. |

| Constraint Language | Natural Language |
|---|---|
| prompt(0, a sheep)
type(0,sheep)
prompt(1, a dog)
type(1,dog)
prompt(2, a giraffe)
type(2,giraffe)
left(0,1,200)
narrower(1,2,100)
below(1,0,200) | A sheep, a dog, and a giraffe.
The sheep is left of the dog.
The dog is narrower than the giraffe.
The dog is below the sheep. |
| prompt(0, a flying pterodactyl)
type(0,giraffe)
prompt(1, an elephant)
type(1,elephant)
below(1,0,500) | Two people flying a kite
were already in the image.
An elephant near the bottom of the
image and a pterodactyl flying above. |
| Detected obj 0 (giraffe)
Detected obj 1 (giraffe)
—
prompt(2, a giraffe)
type(2,giraffe)
prompt(3, a giraffe)
type(3,giraffe)
below(2,0,100)
below(3,2,50)
narrower(2,0)
narrower(3,0)
left(2,0,80)
right(3,0,80) | Two Giraffes were already in the image.
Another two were placed between and left of them. |
| Detected obj 0 (dog)
Detected obj 1 (horse)
—
prompt(2, a sheep)
type(2, sheep)
prompt(3, a brown dog)
type(3, dog)
above(2,1,100)
above(3,1,100)
left(2,0,50)
right(3,1,200) | A dog and horse were already in the image.
A sheep and a brown dog are added above the horse.
The sheep is left of the horse
and the brown dog is right of the horse. |

## D  ARCHITECTURES AND HYPERPARAMETERS

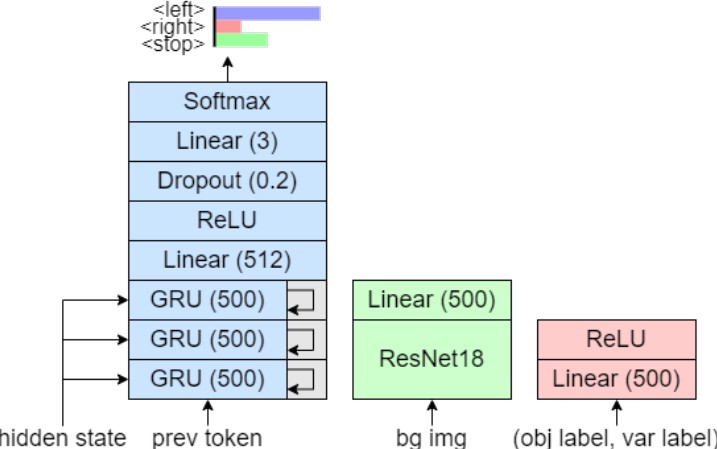

Figure 6: The recurrent network (left), scene encoder (center), and variable encoder (right) of the spatial reasoning net. These networks together allow our method to make sequential decisions on the position of objects in the scene.

## E  GAN + CVX BASELINE

The GAN + CVX approach relies on the recent ability to embed convex optimization solvers into neural layers in such a way that gradients can flow through. In short, this method is accomplished by creating a GAN which outputs a set of parabolas – one for each positional variable – such that the neural net's "best guess" for a location is at the vertex. Then the convex optimizer finds the optimal value of the sum of parabolas subject to constraints.

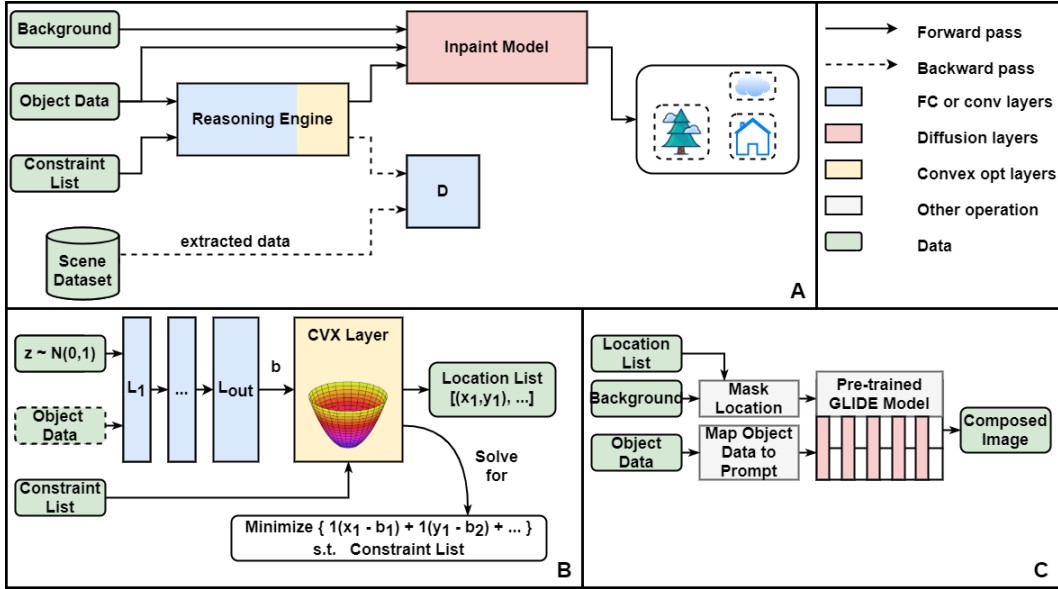

Figure 7: GAN + CVX pipeline. **A)** The full pipeline as a unified system. Three subnetworks are used. A GAN which proposes a list of positions given object data and a constraint list, a pre-trained inpainting model capable of adding the given objects to a background scene, and an adversarial discriminator. **B)** The architecture of the GAN. **C)** The inpainting model.

# F SYNTHETIC DATASETS

Two random functions were used to compose the synthetic datasets. $\mathrm{rnd}(j, k)$ randomly generates an integer between $j$ and $k$ from a normal distribution with mean $j + \frac{k-j}{2}$ and standard deviation $\frac{k-j}{12}$. Selected values are rounded, and constrained between j and k. For example, if object 1's $x$ value is drawn from $\mathrm{rnd}(1, 500)$, then the mean of $x$ will be 250.5 and the standard deviation will be 41.583. $\mathrm{uni}(j, k)$ does the same with a uniform distribution between $j$ and $k$.

## F.1 DS_BASIC

| example prompt | x | y | width (w) | height (h) |
|---|---|---|---|---|
| "a white wolf" | rnd(1,500) | uni(400, 550) | rnd(192,256) | rnd(128,256) |
| "a black wolf" | rnd(500,1000) | uni(400, 550) | rnd(192,256) | rnd(128,256) |

Checked preferences:

- $1 \leq x_{o1} \leq 500$
- $500 \leq x_{o2} \leq 1000$

## F.2 DS_TIGHT

| example prompt | x | y | width (w) | height (h) |
|---|---|---|---|---|
| "a pyramid" | rnd(1,1000) | rnd(300,700) | rnd(220,256) | rnd(120,150) |
| "a cube" | rnd(1,1000) | rnd(300,700) | rnd(120,150) | rnd(120,150) |
| "a sphere" | rnd(1,1000) | rnd(300,700) | rnd(120,150) | rnd(220,256) |

Checked preferences:

- $220 \leq w_{o1} \leq 256$
- $120 \leq w_{o2} \leq 150$
- $120 \leq w_{o3} \leq 150$
- $120 \leq h_{o1} \leq 150$
- $120 \leq h_{o2} \leq 150$
- $220 \leq h_{o3} \leq 256$

### F.3  DS_COMPLEX

| example prompt | x | y |
|---|---|---|
| "a white horse" | rnd(1,1050) | rnd(375,565) |
| "a brown horse" | rnd(1,1050) | rnd($y_{o1} - 10, y_{o1} + 10$) |
| "a fantasy tower" | rnd(1,900) | rnd(1,144) |
| "A pile of colorful cubes" | rnd(1,1050) | rnd(400,665) |
| | **width (w)** | **height (h)** |
| "a white horse" | rnd(64, 128) | rnd($w_{o1} \times 1.5, 200$) |
| "a brown horse" | rnd(64, 128) | rnd($w_{o2} \times 1.5, 200$) |
| "a fantasy tower" | rnd(64, 128) | rnd($w_{o3} \times 2, w_{o3} \times 2$) |
| "A pile of colorful cubes" | rnd(64, 128) | rnd($w_{o4} - 10, w_{o4} + 10$) |

Checked preferences:

- $1 \leq x_{o1} \leq 1050$
- $1 \leq x_{o2} \leq 1050$
- $1 \leq x_{o3} \leq 900$
- $1 \leq x_{o4} \leq 1050$
- $375 \leq y_{o1} \leq 565$
- $y_{o1} - 10 \leq y_{o2} \leq y_{o1} + 10$
- $1 \leq y_{o3} \leq 144$
- $400 \leq y_{o4} \leq 665$
- $64 \leq w_{o1} \leq 128$
- $64 \leq w_{o2} \leq 128$
- $64 \leq w_{o3} \leq 128$
- $64 \leq w_{o4} \leq 128$
- $w_{o1} \times 1.5 \leq h_{o1} \leq 200$
- $w_{o2} \times 1.5 \leq h_{o2} \leq 200$
- $w_{o3} \times 2 \leq h_{o3} \leq w_{o3} \times 2$
- $w_{o4} - 10 \leq h_{o4} \leq w_{o4} + 10$

## G    COCO DATASET

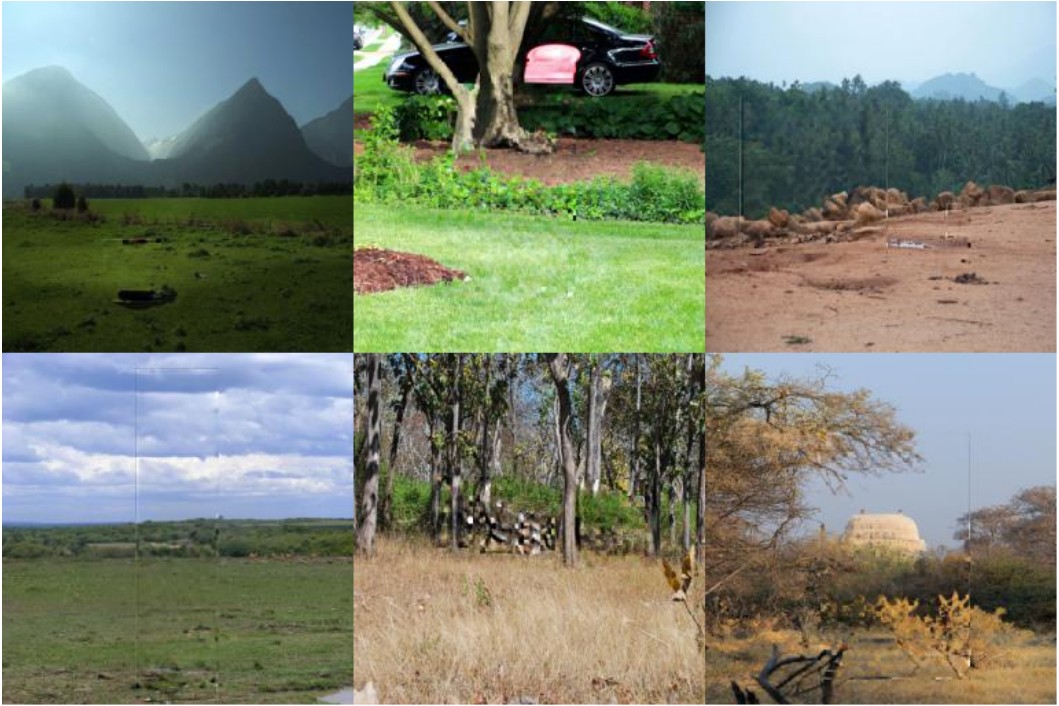

Figure 8: Six examples from the object-removed background testing set. Images like these are fed to the scene encoder for feature extraction before *iterative refinement* begins.

We use the COCO 2017 Detection dataset for training and testing SPREN. During training, SPREN requires access to background images with known ground truth object locations. As COCO does not supply background images without objects in the way. We get around this by utilizing our visual element generator to "paint over" objects to fit the background. An empty prompt is given to the generator, and the object bounding box is covered by a mask, so the generator has no context as to what the object is. This produces object-free backgrounds, or backgrounds that are only stripped of certain objects. We create a background dataset from COCO training data composed of the 10,674 images which contain a dog, horse, sheep, cow, elephant, bear, or giraffe. This dataset is checked manually, but due to its size, minor mistakes may be found in it. For testing, a background dataset of size 20 was acquired in the same way, but was checked more carefully due to its manageable size. Both datasets will be available at the time of publication.

