# OpenReview forum: "Spatial Reasoning Network for Zero-shot Constrained Scene Generation"
_ICLR.cc/2023/Conference — Submitted to ICLR 2023_

### Official Review · Reviewer_4LXR · 2022-10-23

**Confidence:** 4
**Correctness:** 3
**Technical Novelty And Significance:** 3
**Empirical Novelty And Significance:** 3
**Recommendation:** 5

**Clarity, Quality, Novelty And Reproducibility:**

This paper is well-organized and clearly introduce the motivation of proposed method. Author focus on the task of Constrained Scene Generation, as well as the zero-shot and object-aware variants. They design a Spatial Reasoning Network to combine with the off-the-shelf image generation model for enhancing the ability of high-level spatial reasoning, which could be easy to make reproduction. Though with a good motivation and framework, the number of experiments are far from satisfaction, which are lack of detailed analysis for some hyper-parameters and generalization.

**Strength And Weaknesses:**

Strength
1)The greatest strength of this paper is that author propose a plug and play module to enhance the off-the-shelf image generation networks with spatial reasoning ability. Meanwhile, it also could handle zero-shot and object-aware CSG without any fine-tuning, which cost little to increase the quality of generated images. Thus, this work has well practicability and worth to generalize.
2)Besides, author refer to the psychophysiological studies and provide a theoretical explanation that why the cognition needs to combine the spatial constraints, which represented in propositional logic to perform the iterative decision. In this case, the model could explicitly show the reasoning process, which has good interpretability.
3)In experiments, author also verified the effectiveness of spatial reasoning module to stepwise discriminant analysis and evaluated the quality of generated constrained scenes.

Weaknesses
1)Although the proposed spatial reasoning network is effective alone, the quality of generated images still relies heavily on the image generation network, which is implemented by off-the-shelf framework GLIDE. To verity the generalization of spatial reasoning network, I think author should further explore different frameworks of image generation.
2)According to my understanding, the visual element generation module needs to be sequentially called on each box from the blueprint. When the spatial reasoning module provide all the blueprint, image generation network could also directly inpaints all the masked area while keep the global semantic and spatial information. So, I think this sequential manner maybe a little redundant.
3)In section 3.3, author choose 20% as the stop time, but it is not discussed in the experiment. I suggest to add relevant comparative experiments.
4)In practical application, some large-scale pre-trained models could directly realize the similar function for the task of CSG, even under the condition of zero-shot. I think the quality of generated images maybe better than the cases you present in experiments. Meanwhile, the propositional logic mentioned in this paper also requires additional manage. So, how do you view this problem?

**Summary Of The Paper:**

This paper focus on a generative problem of Constrained Scene Generation (CSG) under the condition of zero-shot and object-aware. Inspired by psychophysiological studies, author propose a Spatial Reasoning Network (SPREN) to imitate the multiple systems of reasoning and memory in human beings. Specifically, the framework integrated the state-of-the-art image generation network with the proposed spatial reasoning module to generate image satisfying complex spatial constraints. Besides, they design a forward-checking strategy to ensure the model could handle the zero-shot and object-aware CSG without any fine-tuning. Experiments indicate that SPREN could generate images specifying constrained conditions and also verify the effectiveness in zero-shot and object-aware variants of this task.

**Summary Of The Review:**

As mentioned above, the highlight of this paper is that author proposed a a plug and play module to enhance the off-the-shelf image generation networks with spatial reasoning ability, which meanwhile equipped with a good interpretation from the view of psychophysiological studies. But this work pointed out the key challenge of the spatial reasoning in CSG is vital for future studies. The whole framework is easy to implement and has well practicability, but the experiments is not enough to support its generalizability. Some experiments result still need further analysis.

---

### Official Review · Reviewer_wJp9 · 2022-10-24

**Confidence:** 5
**Correctness:** 3
**Technical Novelty And Significance:** 1
**Empirical Novelty And Significance:** 2
**Recommendation:** 1

**Clarity, Quality, Novelty And Reproducibility:**

The paper is clear and easy to understand. The quality and organization of the writing need to be improved. The novelty is limited (please refer to weaknesses).

It is possible to reproduce the paper.

**Strength And Weaknesses:**

Strengths:
1. The paper is well motivated.
2. The idea of constrained image generation with spatial reasoning is interesting.
3. The paper is easy to follow.

Weaknesses:
1. The experiments are very limited and use uncommon metrics to measure the image generation quality (there is no FID, Inception Score, Precision and Recall, or user study).

2. The literature review is very limited and mostly covers the earlier works on the topic and very few recent works.

3. The novelty is limited. The paper re-introduces existing concepts without referring to them. The proposed blueprints are the same as scene layouts commonly used in image generation literature. Also, the bounding box prediction network is not new. There are a lot of works on the topic of image generation and manipulation using scene graphs and semantic scene generation that are not covered in this work. E.g. sg2im [1] already predicts bounding boxes from a scene graph (similar to the constraints in this work), constructs a scene layout (blueprints), and generates an image conditioned on this. The image manipulation part, which performs addition / deletion / changes to the objects has been explored in SIMSG [2]. [3,4] already learn spatial reasoning based on scene graphs. [5] generates images from scene layouts.

[1] Johnson et al. "Image generation from scene graphs." CVPR 2018.
[2] Dhamo et al. "Semantic image manipulation using scene graphs." CVPR 2020.
[3] Zareian, Alireza, et al. "Learning visual commonsense for robust scene graph generation." ECCV 2020.
[4] Garg et al. "Unconditional scene graph generation." ICCV 2021.
[5] Zhao et al. "Image generation from layout." CVPR 2019.

4. The paper needs better organization. There is too much repetition of text in introduction and methodology that leaves a small space to the experiments which are an essential part of a research paper.

5. The evaluation metrics are not explained. What are pref score, constraint sat, obj sat, pos sat?

**Summary Of The Paper:**

This work proposes a model for spatial reasoning using RNNs for constrained image generation. They compare their method to two baselines and show improved performance on some metrics.

**Summary Of The Review:**

Overall, the paper is low on standards such as evaluation and novelty.

---

### Official Review · Reviewer_9XhE · 2022-10-25

**Confidence:** 4
**Correctness:** 3
**Technical Novelty And Significance:** 2
**Empirical Novelty And Significance:** 2
**Recommendation:** 3

**Clarity, Quality, Novelty And Reproducibility:**

### Quality
OK, but can be improved. Both the justifications of design choices and the quality of results are lacking.

### Clarify
Mostly clear, but many missing details, some wrong terms (e.g. looks like its a recurrent NN not a recursive one), and some descriptions that are unnecessarily convoluted (e.g. S1/S2 cognition, Problem 1/2).

### Originality
The neurally-guided CSP is a nice idea for image generation, but other than that, there isn't much novelty in this submission, with most modules being largely borrowed from existing works.

**Strength And Weaknesses:**

# Strength:
- Using neural nets to guide the solution of CSP is an appropriate solution for this task.
- Output satisfy the constraints well.

# Weaknesses:
### Design choices
In general, there is very little justification for the design choices, to list a few:
- What's the motivation for using this binary partitioning scheme to infer the spatial location of objects. What's the advantage of this over, say, predicting a mixture of gaussians / a distribution over discretized coordinates / etc.?
- Is it reasonable to predict x and y separately? I assume they are highly correlated.
- Is it sufficient to condition the the spatial reasoning module on the "object identifier"? Does this identifier provide enough information to determine the location of objects?
- How is the visual element generation module implemented and how to ensure the generated image patch blends well into the background image?
### Evaluation & Quality of results
The evaluation performed in this submission is largely insufficient:
- Evaluation focused on the amount of constraints satisfied. However, amount of constraints satisfied is not the only thing one cares about when evaluating these images. As the authors acknowledge themselves, visual quality is also very important. No evaluation on the visual quality of the outputs is provided, and, judging from the few qualitative examples provided, these image look appear to be very unrealistic , with a lot of problem with perspectives, illumination, and plausibility of layout in general.
- The comparison conducted is not fair. The proposed method is designed solely around constraint satisfaction and has access to constraint formulations which are not available to the baseline. The baseline is a general purpose framework that accounts for much more than spatial constraints. I expect to see baselines that specifically focuses on constraints, this can be simple baselines such as a pure CSP-based algorithm without the neural guide part.
- I think it's probably more appropriate to evaluate the proposed method on datasets that focused more on object relations and layouts, since the focus of this work does not seem to be image realism anyways. Datasets that target compositionality e.g. CLEVR, or more real-world scene datasets e.g. 3D-FRONT, 3DSSG, can be more appropriate here.
- No ablations at all, which leaves me unsure about a lot of things e.g. how much does the RNN help with solving the CSP? Do the RNN really learn proper location priors e.g. lambs are never placed in the sky?

**Summary Of The Paper:**

This paper proposes a method that composites objects onto a background image in a way that satisfies a set of spatial constraints. A neurally-guided search approach is taken, where a recurrent neural network predicts, via binary space partition, a distribution over possible locations of objects, which is used to guide the solution of a constraint satisfaction problem via backtracking. It is shown that the generated composite matches the constraints better than other general purpose image editing methods. No evaluation on image quality/realism is conducted.

**Summary Of The Review:**

While I find the idea of the paper interesting and that there can be some potentials down this direction, ultimately I am underwhelmed by the justification of the design choices, and more importantly, but the low quality of the results and the lack of appropriate evaluations, particularly on image realism.

---

### Author Response · Authors · 2022-11-30
**Reply to reviewers 9XhE, wJp9, and 4LXR**

**Thank you to all three reviewers for their insightful comments**

This rebuttal will cover two general comments made regarding our work's novelty and evaluation. We will also cover a number of specific comments that we hope to shed some light on.

**Insufficient Evaluation**

We recognize that there are several insufficiencies in the experimentation section that we need to improve for future versions of the paper. New datasets, including compositionality datasets such as CLEVR, will be used for evaluation. An ablation study will also be added, showing the image quality produced when no constraints are included. More baselines will also be found for a more fair comparison.

**Novelty**

The novel advantage of our work is not in the ability to generate scene, or in the ability to generate scene layouts. As mentioned by reviewer wJp9, these capabilities already exist. As does the ability to generate from layouts, generate from scene graphs, and even utilize visual common sense. Our work grants 3 advantages that these do not.
We tackle the problem of constrained scene generation, where the system must output a constrained scene from language input (in this case, a structured language of propositional logic).
Our solution offers guarantees about constraint satisfaction, which purely neural methods do not.
Our solution allows for zero-shot constraint customization, which may be seen to some degree in large language-to-image models, they usually fail with large or complex constraint sets, where ours does not.
This all being said, we recognize that we did a poor job comparing to current scene generation literature to show these advantages. We will improve this in the next version of the work.


**Specific Responses**

Reviewer 9XhE: *Is it reasonable to predict x and y separately? I assume they are highly correlated.*

The spatial variables x and y are not predicted separately. The state of our RNN is maintained through an entire scene. However, one is generated before the other. So if we predict x then y, the network will not know the value of y when predicting x, but it will know the value of x when predicting y. This will be addressed in a later version, perhaps with a bi-GRU.

Reviewer 9XhE: *Is it sufficient to condition the the spatial reasoning module on the "object identifier"? Does this identifier provide enough information to determine the location of objects?*

It also has access to background image through the scene encoder.

Reviewer 9XhE: *How is the visual element generation module implemented and how to ensure the generated image patch blends well into the background image?*

As explained in section 3.4, the pre-trained GLIDE model is used as the visual element generation module. This is a diffusion model, and therefore can inpaint objects into an image given the background as context.


Reviewer 4LXR: *Although the proposed spatial reasoning network is effective alone, the quality of generated images still relies heavily on the image generation network, which is implemented by off-the-shelf framework GLIDE. To verity the generalization of spatial reasoning network, I think author should further explore different frameworks of image generation.*

We will address this in the next version of the paper. Especially considering new SOTA inpainting networks have been released recently.

Reviewer 4LXR: *According to my understanding, the visual element generation module needs to be sequentially called on each box from the blueprint. When the spatial reasoning module provide all the blueprint, image generation network could also directly inpaints all the masked area while keep the global semantic and spatial information. So, I think this sequential manner maybe a little redundant.*

This is a good idea, but we were stopped by the limited output size of GLIDE. We wanted our approach to work for exceedingly large images, while GLIDE is only capable of outputting 128x128 images. In future work, we home to make this tradeoff easier by grouping masks that fit into manageable patches and utilizing newer, bigger inpainting modules.

---

### Decision · Program_Chairs · 2023-01-20

**Decision:**

Reject

**Justification For Why Not Higher Score:**

Three reviewers have negative ratings (1, 3 and 5). During the discussion phase, reviewers pointed out the authors' responses could not address the previous concerns.

**Justification For Why Not Lower Score:**

N/A

**Metareview: Summary, Strengths And Weaknesses:**

This paper aims to deal with the zero-shot constrained scene generation problem by developing a spatial reasoning network. The main idea is to add a spatial reasoning module to existing image generation networks. Experiments show that the proposed method could generate images that satisfy spatial constraints.

Overall, this paper is well motivated and clearly written. The idea of incorporating spatial reasoning to constrained image generation is interesting.

However, there are some major concerns about this submission. In particular, experiments are insufficient, due to the lack of dataset, metrics, and ablation studies. The authors acknowledged this issue in their responses. In addition, some design choices are not well justified, and the novelty of this work is not well clarified, as some related work are not discussed and compared. Hopefully these issues could be addressed in the next version of this work.

**Summary Of Ac-Reviewer Meeting:**

N/A